# DCPNet: A Densely Connected Pyramid Network for Monocular Depth Estimation

**DOI:** 10.3390/s21206780

**Published:** 2021-10-13

**Authors:** Zhitong Lai, Rui Tian, Zhiguo Wu, Nannan Ding, Linjian Sun, Yanjie Wang

**Affiliations:** 1Changchun Institute of Optics, Fine Mechanics and Physics, Chinese Academy of Sciences, Changchun 130033, China; laizhitong17@mails.ucas.ac.cn (Z.L.); wuzg@ciomp.ac.cn (Z.W.); dingnannan@ciomp.ac.cn (N.D.); wangyj@ciomp.ac.cn (Y.W.); 2University of the Chinese Academy of Sciences, Beijing 100049, China; 3National Space Science Center, Chinese Academy of Sciences, Beijing 100190, China; sunlinjian@nssc.ac.cn

**Keywords:** monocular depth estimation, pyramid networks, dense connection, feature fusion

## Abstract

Pyramid architecture is a useful strategy to fuse multi-scale features in deep monocular depth estimation approaches. However, most pyramid networks fuse features only within the adjacent stages in a pyramid structure. To take full advantage of the pyramid structure, inspired by the success of DenseNet, this paper presents DCPNet, a densely connected pyramid network that fuses multi-scale features from multiple stages of the pyramid structure. DCPNet not only performs feature fusion between the adjacent stages, but also non-adjacent stages. To fuse these features, we design a simple and effective dense connection module (DCM). In addition, we offer a new consideration of the common upscale operation in our approach. We believe DCPNet offers a more efficient way to fuse features from multiple scales in a pyramid-like network. We perform extensive experiments using both outdoor and indoor benchmark datasets (i.e., the KITTI and the NYU Depth V2 datasets) and DCPNet achieves the state-of-the-art results.

## 1. Introduction

Monocular depth estimation has been a challenging problem in computer vision research area for decades, which has a huge potential for downstream applications such as 3D reconstruction, robotic navigation and autonomous driving, etc.

In the early days, researchers tried to solve this problem by constructing hand-crafted features [1,2] and probabilistic models [3,4]. With the development of convolutional neural networks (CNNs), the main efforts shifted to devising CNN-based models [5,6,7,8]. Monocular depth estimation using CNNs is such a case where the input is a common rgb image and the output is a pixel-wise depth map that has the same resolution as the input. Capturing context information is critical for pixel-wise prediction problems such as monocular depth estimation. To capture more context information, traditional CNN-based methods mainly refer to enlarging receptive fields of convolution operations, such as using large-size kernels, dilated convolution [9], ASPP [10] and DenseASPP [11], etc. Recently, benefitting from the nice capability to capture context information, the attention mechanism has been widely applied to monocular depth estimation [12,13,14]. Combining traditional CNN-based models with attention mechanism hugely advances the ability to capture more context information.

Another solution to capture more context information aims at devising proper overall architecture of CNNs, where the feature pyramid [15,16,17] is a symbolic structure. Just like a pyramid in the real world, a CNN-based pyramid network is considered as an architecture with different floors, where the scale of features increases or decreases following the floor direction. To make it clear in this paper, we call the stages along the vertical direction of a pyramid network the ’floors’ while the stages along the horizontal direction are called `layers’. The main advantage of the pyramid architecture is that it can fuse multi-scale features from different floors effectively, thus capturing more context information. In convolution operations, for the same feature, intuitively, the small-scale one contains more global information, while the large-scale counterpart contains more local details. Therefore, the fusion between the small-scale features and the large-scale features results in more fruitful context information and improves the representative ability of the features, which is the benefit of using the pyramid structures. Based on this motivation, Miangoleh et al. [18] proposed an outstanding method for merging multi-resolution features. However, their method is not an end-to-end method, and extra pretrained models are required.

Many pyramid-based networks have been applied to monocular depth estimation [17,19,20,21]. Although these methods improve performance by utilizing pyramid architectures, they are not able to take full advantage of using the pyramid architecture. In these pyramid networks, the features at the head of every floor start at different scales, and the connections among floors in these pyramid networks are sparse because they just fuse features with different scales between adjacent floors except the last horizontal layer, instead of fusing the features generated in all the previous floors.

To overcome this problem, inspired by DenseNet [22], we propose a densely connected pyramid network named DCPNet for monocular depth estimation. DCPNet is able to fuse features not only between adjacent floors, but also among non-adjacent floors in a pyramid network, forming a dense frame compared with traditional pyramid networks. Moreover, we design a dense connection module (DCM) to fuse features from different floors effectively, which is an indispensable part of our network. The skip connections in DenseNet are conducted only in the horizontal direction, but in DCPNet they are conducted in both horizontal and vertical directions. We train and test our model on the KITTI dataset and the NYU Depth V2 dataset, which demonstrates that our method achieves state-of-the-art results. Figure 1 shows the examples generated by the proposed method.

The main contributions of our work are summarized as follows:We proposed a novel pyramid network where the connections are much denser than traditional pyramid networks, which is effective for monocular depth estimation.To fuse features among different floors, we designed a dense connection module (DCM), which is simple and effective.We conducted various experiments on both outdoor and indoor datasets (i.e., the KITTI and the NYU Depth V2 datasets). The results demonstrate that the proposed network achieves the state-of-art results on the two datasets.Furthermore, we analyzed two configurations of the common upscale operation, which offers a new direction for us in devising a CNN model.

The rest of this paper is organized as follows. In Section 2, we present a brief survey of related work. In Section 3, we describe the proposed method in detail. Then, experimental results on benchmark datastes and ablation study are provided in Section 4. In Section 5, a brief discussion about our method is put forward. In Section 6, we conclude the paper and provide an outlook for further study.

## 2. Related Work

### 2.1. Monocular Depth Estimation

Extracting the priors of an image is a vital issue in monocular depth estimation. Before the booming of deep neural networks (DNNs), the mainstream method to solve the monocular depth estimation task was utilizing hand-crafted priors [1,2,25] or combining them with probabilistic models [3,4]. With the coming of the DNN era, the procedure of extracting useful features has been conducted by DNN-based models automatically. Many DNN models have been applied to monocular depth estimation tasks. For example, Xu et al. [26] proposed a DNN model combied with conditional random fields (CRFs) for predicting depth images from a single input. Hao et al. [27] proposed an end-to-end DNN model, in which a novel dense feature extractor and a depth map generator were presented. Ye et al. [28] proposed a dual-branch network to preserve the spatial and context information of features for monocular depth estimation task. According to the training manners, DNN models can be sorted into three types: supervised [5,7,20,21], unsupervised [29,30,31,32] and semi-supervised [33,34,35]. For monocular depth estimation, supervised fashion requires a large amount of ground truth depth maps, but unsupervised fashion overcomes this drawback with stereo matching [29,36] or monocular sequences [31,37]. However, due to the lack of ground truth, the accuracies of the unsupervised methods still have a gap with the supervised ones [38]. Hence, semi-supervised fashion begins to attract our attention, which is a fashion combining a small amount of labeled ground truth with a large amount of relatively cheap unlabeled data to improve depth map prediction performance. Though semi-supervised fashion is a promising method to resolve monocular depth estimation problems, supervised fashion is still valuable for us to research due to its higher accuracy.

Encoder–decoder is a standard frame among most DNN-based models for monocular depth estimation, in which the encoder plays a role of extracting prior features of the input color image while reducing the number of parameters to learn. Because the encoders are usually formed by traditional backbones such as VGG [39], ResNet [40], ResNeXt [41] or DenseNet [22], many efforts have been made in devising effective decoders for monocular depth estimation models [20,42,43]. In [42], the encoder is a common backbone, but the decoder contains a depth prediction stream and a semantic labeling stream. The encoders in [20,43] are both common backbones, but the decoder in [20] is a pyramid structure with a local planar guidance module, while the decoder of [43] is combined with the wavelet decomposition tech. Capturing the context information of the features as much as possible is the main research direction in devising a decoder. Because of the outstanding performance of attention mechanism in capturing long-range context information, self-attention and Transformer-style attention become the popular accessories for DNN-based models [12,44,45]. In [12], the authors presented a novel attention mechanism called depth-attention volume (DAV) to capture non-local depth dependencies between coplanar points to leverage monocular depth estimation. Moreover, [44] embedded Transformer into the ResNet [40] backbone and presented a novel unified attention gate structure with independent channel-wise and spatial-wise attention to model long-range global relationships. The authors of [45] proposed a self-supervised DNN model utilizing relational self-attention for jointly learning depth and camera egomotion. Besides attention mechanism, pyramid architecture is also a popular strategy to capture more abundant context information [15,16,17].

In our work, we adopt an encoder–decoder frame for our model, in which the encoder part is a traditional backbone and the decoder is a pyramid-like structure.

### 2.2. Pyramid Netwoks

The pyramid structure is a hierarchical structure that can preserve multi-scale features including both the local details and the global arrangement information. For pixel-wise prediction tasks such as semantic segmation [46,47,48,49,50,51], object detection [52,53,54,55,56,57] and depth map estimation [20,21,58,59,60,61], fusing features in different types and scales can be very helpful for DNN models to learn more fruitful information and then gain better performance.

For semantic segmentation, Seferbekov et al. [46] proposed a feature pyramid network with a downscale pyramid and an upscale pyramid. The features in the former pyramid are added to features in the latter pyramid; then, the added features are concatenated and upsampled to the original image size for the prediction task. Similarly, Chen et al. [47] proposed a feature residual pyramid network (RPNet) where multi-scale residual blocks in the downscale pyramid are preserved to be fused with features in the upscale pyramid. Feng et al. [48] proposed a context pyramid fusion network (CPFNet) for medical image segmentation, which also contains a downscale pyramid and an upscale pyramid. The difference is that before fusing with the upscale pyramid, the intermediate features preserved in the downscale pyramid are fed into global pyramid guidance modules, and the dense feature extracted by the downscale pyramid is fed into a scale-aware pyramid fusion module. Nie et al. [49] introduced a novel bidirectional pyramid network (BPNet) with a downscaling feature flow and a high-resolution maintaining feature flow in which the former flow propagates semantics to the latter flow, while the latter flow passes high-resolution information to the former. Shamsolmoali et al. [50] introduced an adversarial spatial pyramid network (ASPN) in which a pyramid network is incorporated into the Generator of an adversarial network to facilitate domain adaption. Zhang et al. [51] proposed a gated pyramid network (GPNet), in which the features in backbone pyramid are sent to a gated pyramid module, then the output is sent to three cross-layer attention modules for further prediction.

For object detection, Ma et al. [54] proposed a dual refinement feature pyramid network (DRFPN), in which the features in the backbone downscale pyramid are fused with features in the upscale pyramid by spatial refinement blocks, then the generated features are further fused with features in the cascaded downscale pyramid by channel refinement blocks. On the other hand, in [55], the feature fusions between two pyramid directions are completed by attention modules. Wang et al. [53] presented an implicit feature pyramid network (i-FPN), different from explicit feature pyramids as seen in [48,49,54,55], where the backbone pyramid features in i-FPN are fed into a nonlinear transform module as a whole. Liang et al. [57] proposed a novel Mixture Feature Pyramid Network (MFPN) that is integrated by three different feature pyramid networks, which performs differently for different object sizes. Kim et al. [62] proposed a Parallel Feature Pyramid Network (PFPNet), the decoder of which contains three parallel feature rows in different scales; after a bottleneck in every row, the features from three rows are aggregated by three multi-scale context aggregation modules, and the aggregated features are then sent to prediction subnets. Zhao et al. [56] proposed an attention receptive pyramid network (ARPN) for ship detection in SAR images. What special in this network is that the feature fusions between downscale pyramid and upscale pyramid are conducted by three attention receptive blocks. Xin et al. [52] proposed a Reverse Densely Connected Feature Pyramid Network (Rev-Dense FPN), which is very similar to ours, except that the dense connections in Rev-Dense FPN are conducted in just one layer.

To fuse multi-scale features in monocular depth estimation task effectively, Lee et al. [20] presented a pyramid network with a local planar guidance module that highly fused the features from different scales and then obtained a considerable advance in predicting monocular depth. However, extra focal data were required in this method. Liu et al. [21] proposed a multi-scale residual pyramid attention network (MRPAN) that combined pyramid architecture with attention mechanism, which captured the spatial context information and scale context information adaptively. Deng et al. [59] proposed a Fractal Pyramid Networks (PFNs), in which the feature pyramid consists of initial private features and shared features, and finally the two type features are fused to get the depth prediction. Chen et al. [17] proposed a Structure-Aware Residual Pyramid Network (SARPN), in which the backbone features are sent to five adaptive dense feature fusion modules to form a fused feature pyramid, and finally, the fused feature pyramid is fed to the residual pyramid decoder to obtain predicted depth map. Kaushik et al. [60] proposed a pyramid network for self-supervised monocular depth estimation, where the intermediate features in the backbone are sent to four fusion blocks applying CoordConv solution [63] to create a new pyramid sub-network to predict depth map. Poggi et al. [61] proposed a Pyramidal Depth Network (PyD-Net) in which a multi-level feature pyramid is created by the encoder; at each level, a shallow network infers depth at specific resolution, and the inferred depths in lower levels are then upsampled to the above levels to refine estimation. Xu et al. [58] proposed a pyramid network that contains two feature extractors: one is for the full resolution input, and the other one is for the half-resolution input. The two extracted features are then fused by a multi-scale feature fusion dense pyramid, and the motivation of this work is similar as ours.

However, in these methods, apart from [52], the connections to fuse features in different floors of the pyramid structure were sparse. Although the dense connection scheme in [52] exists, the learnable layers on every floor are too shallow. To take full advantage of the pyramid architectures, we devise a densely connected pyramid network without too many complicated tricks.

## 3. Method

In this section, we first descibe the architecture of our proposed network, then introduce the details of dense connection module (DCM). Finally, we introduce the loss function for the training phase.

### 3.1. Network Architecture

An overview of the proposed network is illustrated in Figure 2, which is an encoder–decoder framework as a whole. The encoder part is a traditional pre-trained backbone such as ResNet [40], ResNeXt [41], DenseNet [22], etc. We will implement experiments for different encoders in our network in Section 4. The input color image is highly compressed as a very dense feature (i.e., S/32 feature block in the illustration) through the encoder part, which contains a large amount of deeply stacked convolution blocks. In the encoder process, the intermediate features with the size of S/2, S/4, S/8 and S/16 are preserved to be connected with the decoder part.

The proposed decoder is a pyramid-like structure which contains six floors (i.e., $F_{1}˜F_{6}$ from top to bottom). Each floor contains six layers (i.e., $L_{1}˜L_{6}$ from left to right). In the first floor ($F_{1}$), the dense feature is altered to be the same as the “S/16” feature with respect to the tensor shape through an upscale block. The upscale block is a sequence of a convolution operation and a nearest upsample operation, where the former changes the channel of the feature while maintaining the size of the feature, and the latter enlarges the size of the feature. We will analyze two configurations of the upscale block with respect to the order of the two operations, which will be described in Section 4. The upscale blocks enlarge the size of the features by a ratio of 2, except for the first ones on every floor. The upscale ratios of the first upscale block in floors from $F_{1}˜F_{6}$ are 2, 2, 4, 8, 16 and 32, respectively. The features generated by the upscale operations are preserved to be fused with features in lower floors.

After the upscale operation, a dense connection module (DCM) is used to fuse features on higher floors with features on the current floor. Note that related features on the first floor ($F_{1}$) are fused with features generated from the encoder part. The details of DCM will be described in Section 3.2.

### 3.2. Dense Connection Module

Figure 3 shows the dense connection module. In this illustration, $f_{i,j}$ denotes one of the features in floor *i* as well as layer *j*, which is obtained from an adjacent upscale block and will be sent to next dense connection module. In the DCM, feature $f_{i,j}$ is concatenated with features obtained from DCMs at the same layer *j* from higher floors, then a convolution operation and a sigmoid activation function are used. Note that the convolution operation is just used to alter the channel number of the feature instead of changing the size. The process can be expressed by the following formula:(1)$${\tilde{f}}_{i,j}=Sigmoid\left(Conv\left(\left[f_{i,j},f_{i+1,j},f_{i+2,j},\ldots ,f_{i+n,j}\right]\right)\right)$$
where [•] denotes concatenation operation. ${\tilde{f}}_{i,j}$ indicates the new feature through a DCM, whose shape is the same as feature $f_{i,j}$.

The output of a single DCM is then conveyed to a new upscale block (if it exists), that is
(2)$$f_{i,j+1}=Upscale\left({\tilde{f}}_{i,j}\right)$$

We assume that i=j=6, which means $f_{i,j}=D_{6}$, illustrated in Figure 2. Then, the output of the whole dense pyramid network is
(3)$$\tilde{D}=Sigmoid\left(Conv\left(\left[D_{6},D_{5},D_{4},D_{3},D_{2},D_{1}\right]\right)\right)\times d_{max}$$
where $\tilde{D}$ is the final result predicted from the input color image, and $D_{1}˜D_{6}$ are depth maps generated in every floor, whose shape is the same as $\tilde{D}$. $d_{max}$ denotes the capped maximum depth value, which is 80 for the KITTI dataset, while it is 10 for the NYU Depth V2 dataset.

The pseudocode in Algorithm 1 shows the details of our method, where $P={\left\{P_{k}\right\}}_{k=1}^{n-1}$ denotes the features extracted from encoder pyramid, and $P_{1}$ is the dense feature. *m* and *n* indicate the number of floors and layers, respectively.

### 3.3. Loss Function

We utilize a scale-invariant loss to train our network. Scale-invariant loss is proposed by Eigen et al in [7], which is:(4)L(y,y∗)=1N∑idi2−λN2(∑idi)2
where $d_{i}=logy_{i}-log\hat{y_{i}}$ with $y_{i}$ being the ground truth and $\hat{y_{i}}$ being the predicted depth map at pixel *i*. *N* is the total pixels of a depth map. λ is a hyper-parameter and λ∈[0,1]. We set λ=0.85 here.
**Algorithm 1.** Computing depth with DCPNet**Input:** Backbone feature pyramid: $P={\left\{P_{k}\right\}}_{k=1}^{n-1}$**Output:** Depth map at input scale: $\tilde{D}$
  1:**for**(i=1;i≤m;i=i+1)**do**  2:  **for** (j=i;i≤n;j=j+1) **do**  3:   **if** i==1 **then**  4:     **if** j==1 **then**  5:      ${\tilde{f}}_{i,j}=ReLU\left(BN\left(P_{1}\right)\right)$  6:     **end if**  7:     **if** 1<j<n **then**  8:      $f_{i,j}=Upscale\left({\tilde{f}}_{i,j-1}\right)$  9:      ${\tilde{f}}_{i,j}=Sigmoid\left(Conv\left[f_{i,j},P_{j}\right]\right)$  10:     **end if**  11:     **if** j==n **then**  12:      $f_{i,j}=Upscale\left({\tilde{f}}_{i,j-1}\right)$  13:      $D_{i}=Sigmoid\left(Conv\left[f_{i,j}\right]\right)$  14:     **end if**  15:   **end if**  16:   **if** 1<i≤m **then**  17:     **if** j==i **then**  18:      $f_{i,j}=Upscale\left({\tilde{f}}_{1,1}\right)$  19:      ${\tilde{f}}_{i,j}=Sigmoid\left(Conv\left[f_{i,j},{\tilde{f}}_{i-1,j},{\tilde{f}}_{i-2,j}\ldots ,{\tilde{f}}_{1,j}\right]\right)$  20:     **end if**  21:     **if** i<j<n **then**  22:      $f_{i,j}=Upscale\left({\tilde{f}}_{i,j-1}\right)$  23:      ${\tilde{f}}_{i,j}=Sigmoid\left(Conv\left[f_{i,j},{\tilde{f}}_{i-1,j},{\tilde{f}}_{i-2,j}\ldots ,{\tilde{f}}_{1,j}\right]\right)$  24:     **end if**  25:     **if** j==n **then**  26:      $f_{i,j}=Upscale\left({\tilde{f}}_{i,j-1}\right)$  27:      $D_{i}=Sigmoid\left(Conv\left[f_{i,j},D_{i-1},\ldots ,D_{1}\right]\right)$  28:     **end if**  29:   **end if**  30:  **end for**  31:**end for**  32:$\tilde{D}=D_{m}\times d_{max}$  33:**return**$\tilde{D}$

## 4. Experiments

To evaluate the performance of our proposed method, we conduct experiments in various settings on two benchmark datasets, i.e., the KITTI dataset for outdoor scenario and the NYU Depth V2 dataset for indoor scenario. We provide both quantitative results and qualitative results for our method, as well as compare results with other leading monocular depth estimation methods, i.e., [6,7,17,20,21,28,30,44,58,64,65,66,67].

### 4.1. Datasets

The KITTI dataset [23] is a large-scale dataset of outdoor scenes, which contains plenty of color images with matched ground truth depth maps captured by cameras and depth sensors mounted on a driving car. The resolution of the color images and depth maps is typically 375 × 1242. During training, in order to adapt to hierarchical scale operations, we crop the images to 352 × 704 in a random manner. We follow a split strategy proposed by Eigen et al. [7] to form our training set and test set, where 23,488 images from 32 scenes compose the training set while 697 images from the remaining 29 scenes compose the test set. We set a limitation for the maximum depth value of 80 for all the depth maps.

The NYU Depth V2 dataset [24] is an indoor dataset, which consists of 120K pairs of RGB and depth images captured from 464 indoor scenes by using Microsoft Kinect sensor, whose resolution is 480×640. We crop the images to a resolution of 416×544 randomly for training. We adopt Eigen [7] split strategy which uses 36,253 pairs from 249 scenes for training and 654 pairs from 215 scenes for testing. We cap the depth value to 10 for this dataset.

### 4.2. Evaluation Metrics

We follow several standard evaluation metrics from previous works [7,20,29] to evaluate our proposed method:Mean relative error (absrel): 1N∑i=1N∥di^−di∥di;Mean log10 error (log10): 1N∑i=1N∥logdi^−logdi∥;Squared relative error (sqrel): 1N∑i=1N∥di^−di∥2di;Root mean squared error (rms): 1N∑i=1N(di^−di)2;Root mean squared log10 error (logrms): 1N∑i=1N∥logdi^−logdi∥2;Accuracy with threshold τ, i.e., the percantage (%) of $\hat{d_{i}}$ subjecting to $\delta =max\left(\frac{d_{i}}{\hat{d_{i}}},\frac{\hat{d_{i}}}{d_{i}}\right)<1.25^{\tau }$, here, τ∈(1,2,3).

Here, *N* denotes the total number of valid pixels in the ground truth. $d_{i}$ and $\hat{d_{i}}$ indicate ground truth and predicted depth value at pixel *i*, respectively.

### 4.3. Implementation Details

The proposed method is implemented in the open source deep learning framework Pytorch [68]. We use two NVIDIA 3090 GPUs for all experiments in this work, and the memory of each is 24 GB. The number of epochs is set to 50. We set batch size to 8 for most experiments except some in Section 4.6.2. We initialize the weights with Xavier initialization. During training, we choose AdamW [69] as our optimizer, with $\beta_{1}=0.9$, $\beta_{2}=0.999$, $\epsilon =10^{-6}$ for the KITTI dataset and $\epsilon =10^{-3}$ for the NYU Depth V2 dataset. We set the initial learning rate as $10^{-4}$ and follow the polynomial decay strategy proposed in [20]. To make it clear, the major settings are listed in Table 1.

The encoder of the proposed network is a backbone network such as ResNet101 [40], ResNeXt101 [41] and DenseNet161 [22], which is pretrained on ImageNet [70]. We conduct several experiments on two benchamrk datasets to evaluate the effectiveness of the proposed network under different backbones. The activation functions are ReLU and sigmoid after the convolution operations, where the former is used in upscale blocks and the latter is used in DCMs.

To improve training results and avoid overfitting, data augmentation is employed before inputting images into the network, which contains random horizontal flipping, random contrast, brightness and color adjustment with a chance of 50%. Moreover, we also employ random rotation for the two datastes (range in [−1, 1] for the KITTI dataset and [−2.5, 2.5] for the NYU Depth V2 dataset).

### 4.4. Results on the KITTI Dataset

The quantitative results on the KITTI Eigen test split are shown in Table 2, where [20,28,30,44,69,71,72] adopted the same split strategy as our method. It is worth noting that Lee et al. [20] devised a pyramid-like decoder with an encoder the same as ours, ResNeXt101. Our ResNeXt101 performs better than Lee et al. [20] on metrics sqrel, rms and δ<1.25, which demonstrates the effectiveness of our densely connected decoder. Liu et al. [21] also devised a pyramid-like decoder; however, its performance is worse than ours on all seven metrics.

Figure 4 depicts the qualitative results on the KITTI Eiegn test split. We conduct a comparison between Lee et al. [20] and Alhashim et al. [67]. It is obvious that our method is competitive with Lee et al. [20] and much better than Alhashim et al. [21]. Note that Lee et al. [20] employed the same encoder as ours. From Figure 4, we can observe that our ResNeXt101 performs better in predicting pole-like objects, which are smoother and more coherent at vertical direction.

### 4.5. Results on the NYU Depth V2 Dataset

Table 3 shows the quantitative performance of the proposed method with three variants based on different encoders. We compare them with some existing methods that work on the NYU Depth V2 dataset. Among these methods, Hu et al. [64] adopt the DenseNet-161 encoder, which is the same as ours DenseNet161 model. Moreover, Hu et al. [64] presented a multi-scale feature fusion module (MFF), which also utilizes a dense connection mechanism; however, the connections are sparser than our method and conducted just once. Chen et al. [17] embedded MFF with a Fused Feature Pyramid (FFP) to construct the encoder. The decoder in Chen et al. [17] is a pyramid structure as well. Our method performs best on three metrics (i.e., rms, $\delta <1.25^{2}$ and $\delta <1.25^{3}$), which achieves the state of the art. The quantitative comparisons in Table 3 demonstrate the effectiveness of our devised decoder.

Figure 5 illustrates the qualitative comparison results on the NYU Depth V2 datasets. The dense ground truth depth maps are generated by the toolbox provided on the dataset website. We choose our DenseNet161 model for the comparison. Comparing with Hu et al. [64] and Chen et al. [17], our method obtains more details, especially in contents such as shelves and chair legs. This can explain the characteristic of our method in fusing multi-scale global and local features.

### 4.6. Ablation Study

#### 4.6.1. Effect of Connection Density

We implement an ablation study to demonstrate the effect of the dense connection mechanism in our method. Therefore, we set three setups with respect to the connection density. As shown in Figure 6, we devise three variants of the decoder: from left to right, the counterparts are Sparse, Sparser and Sparsest, respectively. For the Sparse setup, we connect features that are just between the adjacent floors; for the Sparser setup, we do the dense connections just in the last layer; while for the Sparsest setup, the connections in the last layer are sparse. Table 4 shows the quantitative results among the three variants. From Table 4, we can observe that the overall performances drop along connection density decreasing. This demonstrates the effectiveness of our densely connected mechanism. Moreover, even the Sparsest setup achieves the state-of-the-art result, which offers further guidance for us in advancing the fundamental pyramid framework.

#### 4.6.2. Effect of Different Upscale Operations

Upscale operation is almost an indispensable module in most CNN-based models, let alone the pyramid networks. Most models conduct upscale operation like in Figure 7a, where a convolution block is followed by an upsample operation, such as in [17,19,20,44]. The function of the convolution block is to alter the channel number of the features while keeing the size, and the upsample operation enlarges the size while not changing the channel number. The sequence of upsample–convolution operation changes the channel number and size of the features simultaneously. However, as found in our work, the configuration in Figure 7b works better than the traditional one depicted in Figure 7a. The two configurations only differ in the order of convolution and upsample operation.

In our work, as shown in Figure 7b, the upsample operation is followed by a convolution block, which is opposite to the configuration in Figure 7a. With the same number of learnable parameters, the computational complexities of the two configurations are different. The computational complexity of ours is $C_{1}C_{2}H_{1}^{2}W_{1}^{2}$, while it is $C_{1}C_{2}H_{2}^{2}W_{2}^{2}$ in the upsample–convolution sequence. It is obvious that the former is smaller than the latter.

We evaluate the two upscale schemes within our network under different encoders and two state-of-the-art works (i.e., BTS [20] and TransDepth [44]) on the KITTI dataset. After training, we test the mean inferring time on the KITTI Eigen test split on the platform Intel i7 9700K CPU. As shown in Table 5, for our method, the two schemes perform similarly, while the convolution-upsample sequence consumes less time in inferring a single input. However, for BTS pair and TransDepth pair, the time consumptions are similar, while the original BTS method performs better than conv_up_BTS method. We infer that this is due to the small number of upscale operations in BTS and TransDepth methods; thus, the sequence order in upscale operations is not sensitive to the overall time consumption. Nevertheless, in TransDepth pair, conv_up_TransDepth performs much better than TransDepth on metric rms, which still offers a new direction for us in devising a CNN-based network, especially for the one containing many upscale operations.

## 5. Discussion

The experimental results demonstrate that our method is effective for monocular depth estimation. Compared with other pyramid-like methods, our proposed method constructs a denser pyramid structure, which can be helpful to fuse multi-scale features. However, a huge memory cost is required for our method since the intermediate features are preserved to be fused with features in lower floors all the time. To alleviate this problem, we try to decrease the floor number of the pyramid structure gradually, while maintaining the dense connections. As shown in Table 6, even F#3-DenseNet161 achieves the state of the art, the performance of which is just slightly worse than our DenseNet161, which further demonstrates the effectiveness of our idea. However, the performances do not degrade along the decreasing of floor numbers, and the reasons behind this are worthy of further research.

## 6. Conclusions

In this work, we proposed a densely connected pyramid network for monocular depth estimation. Unlike the traditional pyramid-structure neural networks, where the feature fusion is just conducted between adjacent floors of the pyramid, we fused features not only in adjacent floors but also in non-adjacent floors by dense connection modules. Furthermore, the upscale operations different from traditional ones are applied in our network, which results in similar performances but is less time-consuming. Experimental results on the KITTI dataset and the NYU Depth V2 dataset show that the proposed method is effective for monocular depth estimation. As our method is constructed by a pure convolution mechanism, we plan to investigate the combination with the trendy attention mechanism as future works.

## Figures and Tables

**Figure 1 sensors-21-06780-f001:**
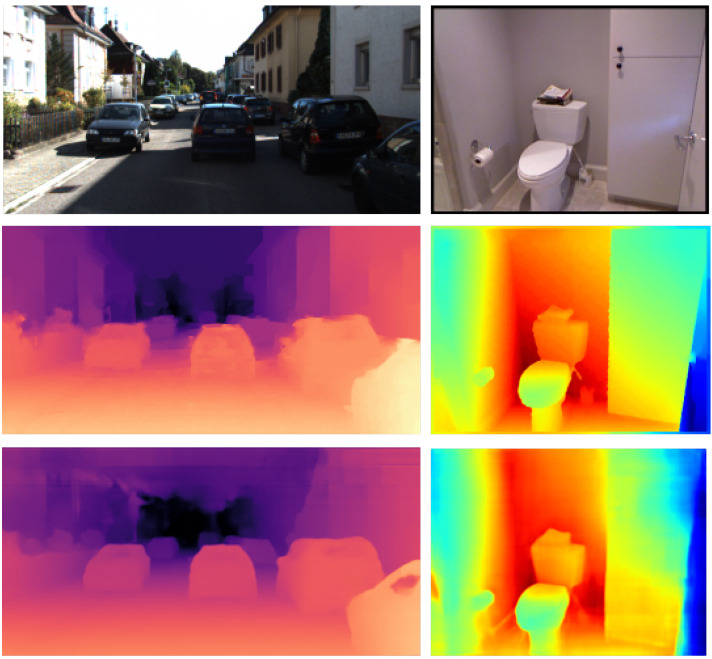
Depth estimation examples. From top to bottom: input color images, ground truths and outputs inferred by the proposed network. Note that the left example is from the KITTI dataset [23], and the right one belongs to the NYU Depth V2 dataset [24].

**Figure 2 sensors-21-06780-f002:**
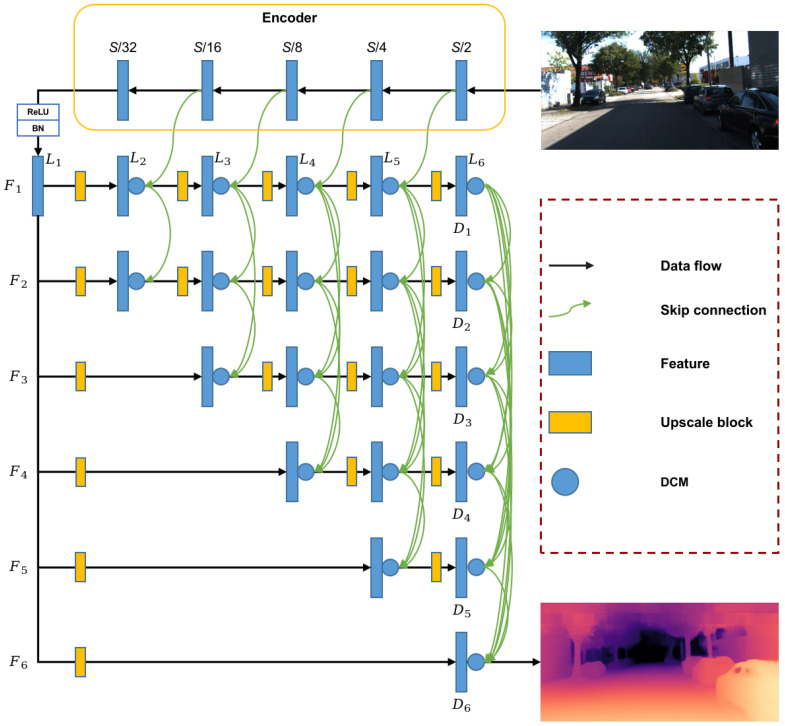
The overall architecture of the proposed network. *S* indicates the spatial resolution of the input color image. $F_{1}˜F_{6}$ denote floors of the pyramid structure. $L_{1}˜L_{6}$ denote layers in every floor, and some layers do not exist on specific floors. The resolution of the features on the last layer of every floor is *S* (i.e., $D_{1}˜D_{6}$, whose resolution is the same as the input image). The blue circles indicate dense connection module (DCM), which will be described in detail in Section 3.2.

**Figure 3 sensors-21-06780-f003:**
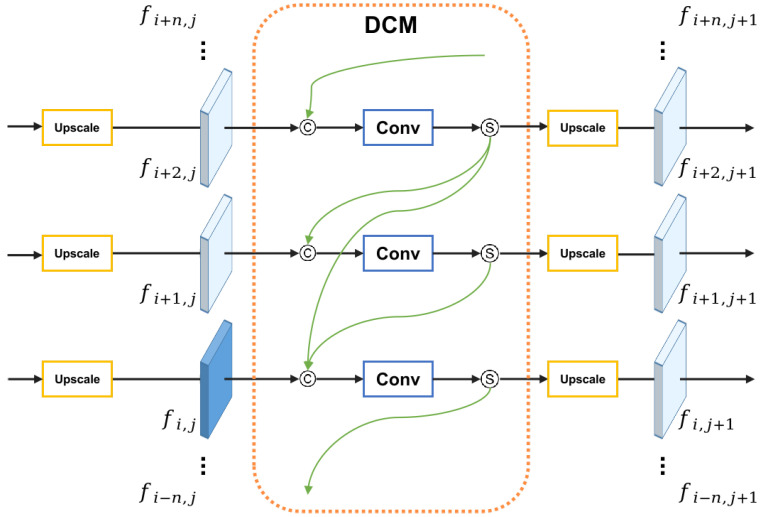
The overview of our presented dense connection module (DCM), where the symbols © and Ⓢ indicate concatenation and sigmoid activation, respectively.

**Figure 4 sensors-21-06780-f004:**
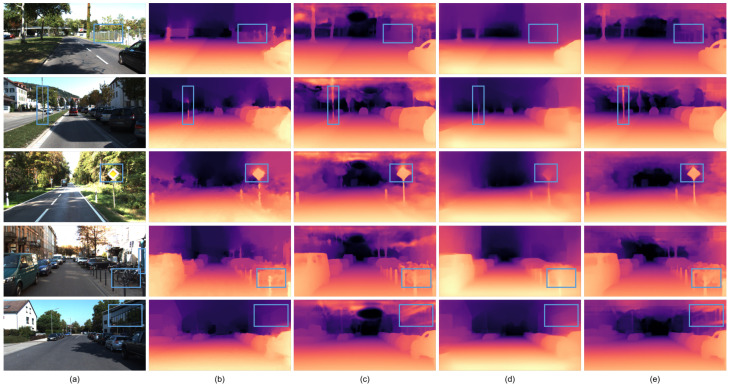
Qualitative examples on the KITTI Eigen test split. (**a**) RGB image; (**b**) ground truth; (**c**) Lee et al. [20]; (**d**) Alhashim et al. [67]; (**e**) Ours—ResNeXt101. The ground truth depth maps are filled based on sparse point clouds utilizing tools provided by the NYU Depth V2 dataset. For better visualization, the values of all the depth maps are logarithmic. Note that the encoders of Lee et al. [20] and ours are both ResNeXt101.

**Figure 5 sensors-21-06780-f005:**
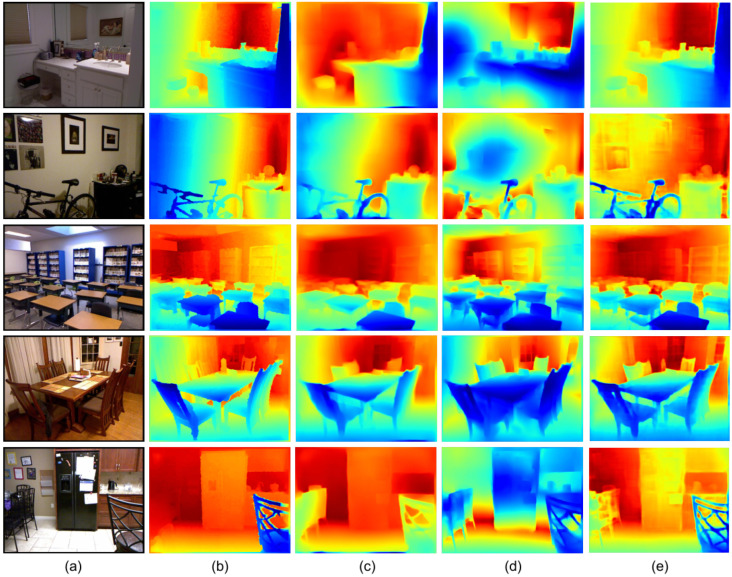
Qualitative examples on the NYU Depth V2 Eigen test split. (**a**) RGB image; (**b**) ground truth; (**c**) Hu et al. [64]; (**d**) Chen et al. [17]; (**e**) ours. From top to bottom, we select five RGB images from five scenes, i.e., bathroom, bedroom, classroom, dining room and kitchen, respectively. Note that the encoder of ours is DenseNet161.

**Figure 6 sensors-21-06780-f006:**
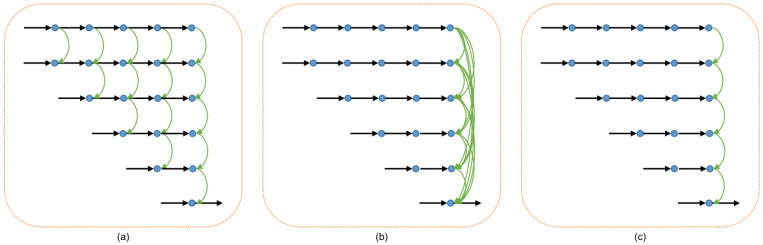
Variants of the decoder architecture according to the connection density. (**a**) Sparse; (**b**) Sparser; (**c**) Sparsest. Here, we omit the feature blocks and upscale blocks depicted in Figure 2 (i.e., blue blocks and yellow blocks).

**Figure 7 sensors-21-06780-f007:**
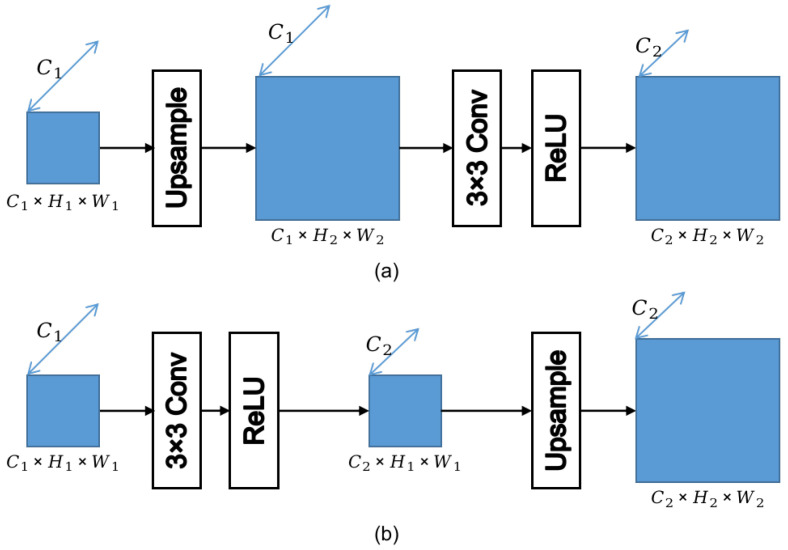
Two configurations of upscale operation. (**a**) upsample–convolution sequence; (**b**) convolution–upsample sequence.

**Table 1 sensors-21-06780-t001:** Training settings on two datasets.

Settings	KITTI	NYU Depth V2
epochs	50	50
batch size	8	8
optimizer	AdamW	AdamW
optimizer ϵ	1 × 10${}^{-6}$	1 × 10${}^{-3}$
input height	352	416
input width	704	544
initial learning rate	1 × 10${}^{-4}$	1 × 10${}^{-4}$
initialization method	Xavier	Xavier

**Table 2 sensors-21-06780-t002:** Quantitative results on the KITTI Eigen test split. The best results of every metric among these works are marked in bold type, while the second best ones are underlined. We employ three encoders for our method, i.e., ResNet101, ResNeXt101 and DenseNet161. Note that the maximum depth of these methods are all set to 80 m. Metrics indicated by ↓: lower is better; metrics indicated by ↑: higher is better.

Method	absrel↓	sqrel↓	rms↓	logrms↓	δ<1.25↑	$\delta <1.25^{2}$↑	$\delta <1.25^{3}$↑
Eiegn et al. [7]	0.190	1.515	7.156	0.270	0.692	0.899	0.967
Liu et al. [6]	0.217	-	7.046	-	0.656	0.881	0.958
Alhashim et al. [67]	0.093	0.589	4.170	-	0.886	0.965	0.986
Fu et al. [65]	0.072	0.307	2.727	0.120	0.932	0.984	0.994
Yin et al. [66]	0.072	-	3.258	0.117	0.938	0.990	0.998
Lee et al. [20]	**0.059**	0.245	2.756	**0.096**	0.956	0.993	0.998
Godard et al. [30]	0.106	0.806	4.530	0.193	0.876	0.958	0.980
Yang et al. [44]	0.064	0.252	2.755	0.098	0.956	**0.994**	**0.999**
Liu et al. [21]	0.111	-	3.514	-	0.878	0.977	0.994
Ye et al. [28]	0.112	-	4.978	0.210	0.842	0.947	0.973
Ours—ResNet101	0.065	0.257	2.779	0.101	0.950	0.992	0.998
Ours—ResNeXt101	0.061	**0.238**	**2.699**	**0.096**	**0.957**	0.993	0.998
Ours—DenseNet161	0.063	0.248	2.762	0.098	0.955	0.993	**0.999**

**Table 3 sensors-21-06780-t003:** The quantitative results on the NYU Depth V2 dataset with Eigen split. Note that the best results of every metric among these works are marked in bold type, while the second best ones are underlined. We employ three encoders for our method, i.e., ResNet101, ResNeXt101 and DenseNet161. Metrics indicated by ↓: lower is better; metrics indicated by ↑: higher is better.

Method	absrel↓	log10↓	rms↓	δ<1.25↑	$\delta <1.25^{2}$↑	$\delta <1.25^{3}$↑
Eiegn et al. [7]	0.215	-	0.907	0.611	0.887	0.971
Liu et al. [6]	0.213	0.087	0.759	0.650	0.906	0.976
Fu et al. [65]	0.115	0.051	0.509	0.828	0.965	0.992
Hu et al. [64]	0.123	0.053	0.544	0.855	0.972	0.993
Yin et al. [66]	0.108	**0.048**	0.416	0.875	0.976	0.994
Chen et al. [17]	0.111	**0.048**	0.514	**0.878**	0.977	0.994
Liu et al. [21]	0.113	0.049	0.525	0.872	0.974	0.993
Ye et al. [28]	-	0.063	0.474	0.784	0.948	0.986
Xu et al. [58]	**0.101**	0.054	0.456	0.823	0.962	0.994
Ours—ResNet101	0.126	0.054	0.433	0.846	0.974	**0.995**
Ours—ResNeXt101	0.117	0.051	0.408	0.865	0.979	**0.995**
Ours—DenseNet161	0.115	0.049	**0.398**	0.873	**0.980**	**0.995**

**Table 4 sensors-21-06780-t004:** Ablation study on the KITTI dataset: performance for different connection densities. All the variants are implemented under DenseNet161 encoder. Metrics indicated by ↓: lower is better; metrics indicated by ↑: higher is better.

Method	absrel↓	sqrel↓	rms↓	logrms↓	δ<1.25↑	$\delta <1.25^{2}$↑	$\delta <1.25^{3}$↑
Ours—DenseNet161	0.063	0.248	2.762	0.098	0.955	0.993	0.999
Sparse	0.061	0.250	2.821	0.098	0.954	0.992	0.998
Sparser	0.062	0.252	2.804	0.098	0.953	0.993	0.998
Sparsest	0.061	0.248	2.815	0.098	0.953	0.993	0.998

**Table 5 sensors-21-06780-t005:** Ablation study of two upscale schemes on the KITTI dataset. Four backbones with different amount of parameters are chosen for our proposed decoder. Here, ours and conv_up denote convolution–upsample sequence while up_conv denotes upsample–convolution sequence. The unit of time is second. Restricted by the GPU memory, the batch sizes are set to 4 for experiments up_conv_DenseNet161, conv_up_TransDepth and ResNeXt50 pair. Metrics indicated by ↓: lower is better; Metrics indicated by ↑: higher is better.

Method	#Params	Time	absrel↓	sqrel↓	rms↓	logrms↓	δ<1.25↑	$\delta <1.25^{2}$↑	$\delta <1.25^{3}$↑
Ours—MobileNetV2	5.7M	0.0270	0.071	0.295	2.978	0.109	0.940	0.991	0.998
up_conv—MobileNetV2	5.7M	0.0338	0.071	0.291	2.971	0.110	0.940	0.991	0.998
Ours—ResNet34	30.8M	0.0235	0.064	0.259	2.787	0.100	0.952	0.993	0.998
up_conv—ResNet34	30.8M	0.0387	0.064	0.259	2.856	0.101	0.951	0.993	0.998
Ours—DenseNet161	63.9M	0.0617	0.063	0.248	2.762	0.098	0.955	0.993	0.999
up_conv—DenseNet161	63.9M	0.1346	0.064	0.252	2.765	0.100	0.951	0.992	0.998
Ours—ResNeXt50	155.2M	0.0741	0.064	0.252	2.793	0.101	0.950	0.992	0.998
up_conv—ResNeXt50	155.2M	0.1370	0.066	0.262	2.768	0.102	0.948	0.992	0.998
conv_up_BTS	47.0M	0.0482	0.065	0.253	2.820	0.100	0.952	0.993	0.999
BTS [20]	47.0M	0.0481	0.060	0.249	2.798	0.096	0.956	0.993	0.998
conv_up_TransDepth	247.4M	0.1703	0.065	0.258	2.738	0.098	0.953	0.993	0.999
TransDepth [44]	247.4M	0.1705	0.064	0.252	2.755	0.098	0.956	0.994	0.999

**Table 6 sensors-21-06780-t006:** The quantitative results of the pyramid structures with different floor numbers on the KITTI dataset. F#5, F#4, F#3 indicate the pyramids with 5 floors, 4 floors and 3 floors, respectively. Note that the floor number of ours—DenseNet161 is 6. Metrics indicated by ↓: lower is better; metrics indicated by ↑: higher is better.

Method	absrel↓	sqrel↓	rms↓	logrms↓	δ<1.25↑	$\delta <1.25^{2}$↑	$\delta <1.25^{3}$↑
Ours—DenseNet161	0.063	0.248	2.762	0.098	0.955	0.993	0.999
F#5—DenseNet161	0.062	0.253	2.793	0.098	0.954	0.992	0.998
F#4—DenseNet161	0.061	0.248	2.789	0.097	0.955	0.993	0.998
F#3—DenseNet161	0.063	0.251	2.765	0.099	0.953	0.993	0.998

## Data Availability

Not applicable.

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
