# Peer review of "DCPNet: A Densely Connected Pyramid Network for Monocular Depth Estimation"

_sensors, 2021, doi:10.3390/s21206780_

Round 1

Reviewer 1 Report

Thanks to the authors for this good work. Monocular Depth Estimation has been an applicable research topic since 2019. This work presents DCPNet, a densely connected pyramid network that fuses multi-scale features from multiple stages of the pyramid structure with the most two popular datasets - KITTI and NYU Depth V2 datasets. The paper is generally good, I only have two suggestions as follows:

  1. Although time is shown in Table 4, but I want to see a more clear comparison of time cost with different cutting edge algorithms. Densly Connected Pyramid is sort of one kind of time-consuming architecture, so a detailed time cost matrix is nesserary.
  2. Since this is a new submission, so I expect some relative 2021 CVPR works to be included as references/baselines. For example:  Boosting Monocular Depth Estimation Models to High-Resolution via Content-Adaptive Multi-Resolution Merging, Knowledge Distillation for Fast and Accurate Monocular Depth Estimation on Mobile Devices and etc. 

Reviewer 2 Report

The authors propose a densely connected pyramid network for monocular depth estimation.

Unfortunately, the paper contains the following problems:

  • In section 2.1, the authors cited a number of papers without explaining how they work. A brief explanation of the structure for each method is necessary so the reader can understand the structure of each method without having to read each cited paper. The parts needing this explanation are the following: “Many DNN models have been applied to monocular depth estimation tasks [25–28,31].”, “many efforts have been made in devising effective decoders for monocular depth estimation models [7,18,43].” and “Because of the outstanding performance of attention mechanism in capturing long-range context information, self-attention and Transformer-style attention become the popular accessories for DNN-based models [11,31,44,45]. Besides attention mechanism, pyramid architecture is also a popular strategy to capture more abundant context information [14–16].”
  • Section 2.2 is very small and needs significant expansion. The authors must include at least 20 existing methods regarding pyramid networks, followed by a brief explanation of each method’s structure.
  • Each figure must be placed after it is cited for the first time.
  • In section 3, the authors must include the pseudocode of their proposed method followed by an explanation.
  • Section 4 is lacking critical information about the initialization and execution of the experiments. A table containing the parameters for each method should be included in the experimental part. This table, followed by an explanation, would give a better understanding of the parameters used.
  • The authors should create a separate discussion section before the conclusions section.

Based on the above, critical information is missing and the manuscript should be reevaluated after undergoing a major revision.

Round 2

Reviewer 2 Report

The authors have addressed all my comments.

This manuscript is a resubmission of an earlier submission. The following is a list of the peer review reports and author responses from that submission.